# Ultrafast Backbone Protonation in Channelrhodopsin-1 Captured by Polarization Resolved Fs Vis-pump—IR-Probe Spectroscopy and Computational Methods

**DOI:** 10.3390/molecules25040848

**Published:** 2020-02-14

**Authors:** Till Stensitzki, Suliman Adam, Ramona Schlesinger, Igor Schapiro, Karsten Heyne

**Affiliations:** 1Institut für Experimentalphysik, Freie Universität Berlin, Arnimallee 14, 1495 Berlin, Germany; tillsten@zedat.fu-berlin.de (T.S.); rschlesinger@zedat.fu-berlin.de (R.S.); 2Fritz Haber Center for Molecular Dynamics Research, Institute of Chemistry, The Hebrew University of Jerusalem, Givat Ram, Jerusalem 9190401, Israel; suliman.adam@mail.huji.ac.il (S.A.); Igor.Schapiro@mail.huji.ac.il (I.S.)

**Keywords:** photoisomerization, retinal, optogenetics, vibrational spectroscopy, QM/MM calculations, protein dynamics, counter-ion, proton transfer, proton back-transfer, channelrhodopsin

## Abstract

Channelrhodopsins (ChR) are light-gated ion-channels heavily used in optogenetics. Upon light excitation an ultrafast all-*trans* to 13-*cis* isomerization of the retinal chromophore takes place. It is still uncertain by what means this reaction leads to further protein changes and channel conductivity. Channelrhodopsin-1 in *Chlamydomonas augustae* exhibits a 100 fs photoisomerization and a protonated counterion complex. By polarization resolved ultrafast spectroscopy in the mid-IR we show that the initial reaction of the retinal is accompanied by changes in the protein backbone and ultrafast protonation changes at the counterion complex comprising Asp299 and Glu169. In combination with homology modelling and quantum mechanics/molecular mechanics (QM/MM) geometry optimization we assign the protonation dynamics to ultrafast deprotonation of Glu169, and transient protonation of the Glu169 backbone, followed by a proton transfer from the backbone to the carboxylate group of Asp299 on a timescale of tens of picoseconds. The second proton transfer is not related to retinal dynamics and reflects pure protein changes in the first photoproduct. We assume these protein dynamics to be the first steps in a cascade of protein-wide changes resulting in channel conductivity.

## 1. Introduction

The discovery of channelrhodopsins [1,2], a light-gated cation channel which can be expressed in animal cells, lead to the invention of optogenetics. This new application field renewed the interest in the photo-physics of rhodopsins, in particular how to tailor optical properties of rhodopsins for applications, i.e., the absorption wavelength of the rhodopsin, and the yield for channel opening. This requires a molecular level understanding of the chromophore–protein interactions. Hence, a growing number of studies investigate the mechanism of the photo-induced reaction, which starts, like in all rhodopsins, with photoisomerization of the retinal chromophore.

How this reaction couples to the protein and transfers the photoisomerization to effective channel opening is still not understood. Upon photoexcitation, it was demonstrated that the protein response in rhodopsins can be as fast as or even faster than retinal isomerization [3,4,5]. Since excitation of rhodopsins is connected with a significant charge translocation along the chromophore, the altered electric field is expected to interact with the protein [6]. In principle, these interactions induce changes which should be observable by femtosecond mid-IR spectroscopy. So far, the assignment of spectral features to specific protein interactions is lacking. Significant changes were observed for the amide I band upon excitation, showing a significant loss of oscillator strength in several rhodopsins, e.g., in channelrhodopsin-2 (*Cr*ChR2) [7,8], less pronounced in bacteriorhodopsin (bR) [9], locked bR [10], and sensory rhodopsin II (*Np*SRII) [9]. Observation of an amide I band change on the ultrashort timescale is surprising, since these features are generally associated with protein structural changes on timescales in the µs to ms. The mechanism which causes the ultrafast amide I change is unknown. Without an understanding of the loss in oscillator strength of the amide I band, the ubiquitous interpretation of amide I changes equaling structural changes is questionable and the picture of the chromophore–protein interaction is incomplete.

A promising candidate for optogenetics is channelrhodopsin-1 from *Chlamydomonas augustae* (*Ca*ChR1) with a red-shifted absorption maximum and a lack of fast inactivation [11]. Up to now, there is no crystal structure of *Ca*ChR1 available and the exact spatial position of individual amino acid residues is unknown. However, *Ca*ChR1 is well studied by steady-state Raman and cryo-FTIR spectroscopy [12,13,14,15]. *Ca*ChR1 exhibits a neutral glutamate in the counterion complex [13], and has an extremely fast 100 fs photoisomerization of the retinal [16,17,18].

In this study, we combine homology modelling and quantum mechanics/molecular mechanics (QM/MM) geometry optimization to obtain the initial state structure of the chromophore and surrounding amino acids (see Figure 1) with polarization resolved femtosecond mid-IR spectroscopy tracing both dynamics and relative orientations of the chromophore and residues of *Ca*ChR1 in real time.

By comparing angles from the structural model with measured relative angles, we are able to single out contributions of individual amino acid groups in the initial ground state, and track their vibrational dynamics upon photoexcitation [19]. From the composed picture we propose a mechanism involving the proton transfer from the carboxylate group of the counterion Glu169 to its backbone peptide group during the isomerization. This proton transfer is followed by a deprotonation of the Glu169 backbone peptide group and protonation of the counterion carboxylate of Asp299.

## 2. Results

### 2.1. Computational Results

Since no structure of *Ca*ChR1 is available yet, we apply homology modelling and QM/MM geometry optimization to obtain a ground-state structural model of *Ca*ChR1. The part of the structural model around the retinal chromophore is depicted in Figure 1. The counterion of the Schiff base comprises Glu169 and Asp299, in direct contact to retinal’s N-H group in the all-*trans* ground state. The S-H group of Cys173 is connected via a hydrogen bond to the backbone C=O group of Glu169. Using this structural model, we calculate the orientation of retinal’s electronic transition dipole moment (tdm) of the S_0_ → S_1_ transition within the protein, and determine the relative angle between this electronic tdm and all vibrational tdms of C=O groups in the model (see Appendix A). As depicted in Figure 1 the electronic tdm is oriented along the retinal polyene chain (blue arrow). The orientation of the vibrational C=O stretching tdms are estimated using simple heuristics [20].

### 2.2. Experimental Results

Figure 2 (top row) shows the isotropic transient spectra for several delay times and an averaged spectrum for parallel and perpendicular polarizations (bottom row). Moreover, regions dominated by retinal modes, which are known from prior resonance Raman studies [12,14,18], are shaded in orange. The data was recorded in two separate runs with different spectral resolution. The C=C stretching mode of retinal absorbs at 1550 cm^−1^ in the ground state and shifts to 1530 cm^−1^ in the photoproduct state. The C=ND stretching mode of the Schiff base absorbs at 1623 cm^−1^ in the ground state and shifts to 1602 cm^−1^ upon photoisomerization. The other signals are assigned to ultrafast protein response (a detailed discussion of the shown transient spectra can be found in Appendix A).

### 2.3. Assignment of Protein Bands

Since it was shown that in *Ca*ChR1 retinal isomerizes with a time constant of 100 fs and relaxes within several picoseconds [16], transient processes on longer timescales reflect protein dynamics. Measurements on retinal analogs showed that no retinal contributions [21] are found in the wavenumber range of 1650 cm^−1^ and 1800 cm^−1^. In Figure 2, a prominent feature is the strong bleaching band at 1661 cm^−1^. As shown in Figure 3, the bleaching signal at 1661 cm^−1^ appears instantaneously. This strong bleaching band has no corresponding positive counterpart, excluding a simple frequency shift, e.g., due to changed electrostatics. Thus, we observe an ultrafast loss of oscillator strength for the band at 1661 cm^−1^. Similar instantaneous bleaching features around 1660 cm^−1^ were observed in CrChR2 [7], in bR [10], and NpSRII [9], and were attributed to an amide I mode. Thus, we assign this band to an amide I mode in *Ca*ChR1. Later, on a timescale of tens of picoseconds the bleaching signal at 1661 cm^–1^ loses intensity, due to the rise of a positive band at 1668 cm^−1^ (see Figure 2 top row). This indicates formation of a new positive band at 1668 cm^−1^.

Additionally, an instantaneous bleaching band is visible at 1690 cm^−1^ in Figure 2, Figure 3 and Figure 4. This signal is likely related to the C=O stretching vibration of a strongly hydrogen bonded carboxyl group [22]. This vibrational band exhibits an instantaneous loss of oscillator strength, probably induced by deprotonation of the carboxyl group. Since in CaChR1 the only protonated carboxylic acid near the retinal is Glu169, we assign this vibrational band to Glu169. The bleaching signal displays a negligible decay on the timescale of 40 ps (Figure 3 and Figure 4). Deprotonation of a carboxylate group generates a COO- group with asymmetric stretching vibration around 1570 cm^−1^. We observe instantaneous positive signals at this frequency position supporting our assignment, but this spectral range is too congested to gain further information.

On a longer timescale, a broad positive band at 1703 cm^−1^ rises within tens of picoseconds. The signal is visible for parallel polarization at 1703 cm^−1^ (Figure 3, black squares) and rises after 10 ps. A clear indication for this band is given by the polarization resolved signals averaged from 23 ps to 50 ps presented in Figure 4. Vibrations in the region from 1760 cm^−1^ to 1700 cm^−1^ are attributed to C=O groups in rhodopsins. Since we found no negative band from 1700 cm^−1^ to 1750 cm^–1^, the only adjacent negative band is located at 1690 cm^−1^. Because this bleaching band shows negligible changes within tens of picoseconds, these two bands represent distinct C=O groups. We assign the band at 1703 cm^−1^ to the protonation of a carboxylate group.

Ogren et al. assigned the vibrational band at 1703 cm^−1^ to Asp299 and proposed proton transfer between the counterion groups Glu169 and Asp299 as part of the initial photoreaction [13]. Hence, we assign the vibrational band at 1703 cm^−1^ to protonation of Asp299. But the open question is, which group is the proton donor for Asp299 on the timescale of tens of picoseconds since Glu169 gets deprotonated earlier.

### 2.4. Spectral Decomposition

We modelled the complete dynamics of the polarization resolved signals from 1750 cm^−1^ to 1655 cm^−1^ with four peaks. Two negative bleaching bands kept constant in time, while the rise of two positive bands is modelled with a single shared exponential. Experimental data and simulated results are presented in Figure 4 for averaged delay times. The thick lines in Figure 4 represent the simulated spectra. The time constant was fitted to 16 ps (2σ: 11–25 ps). At 1661 cm^−1^ we found a relative angle of the amide I bleaching band of 45° (2σ: 42–48°), the negative carboxyl stretching band at 1689 cm^−1^ has a relative angle of 29° (2σ: 23–35°), the rising positive band at 1668 cm^-1^ exhibits an angle of 41° (2σ: 32–49°), and the rising band at 1703 cm^−1^ has a relative angle of 0° (2σ: 0–30°). The 2σ ranges are calculated by exhaustive search via the Python package lmfit [23]. Since, the positive band at 1668 cm^−1^ is adjacent to the negative amide I band at 1661 cm^−1^, and exhibits the same angle within the error margin, we assign the positive band at 1668 cm^−1^ to the same amide I vibration absorbing at 1661 cm^−1^ in the initial state, frequency shifted due to an altered surrounding in the photoproduct state. The angle of the positive band at 1703 cm^−1^ does not match the bleaching band angle of Glu169, corroborating the assignment to Asp299.

The structure of rhodopsins with seven transmembrane helices comprises backbone C=O groups nearly perpendicular to the retinal tdm [24]. Most amide I angles in *Ca*ChR1 are grouped around 70° (see Appendix A). The relative angle of 45° of the amide I bleaching band at 1661 cm^−1^ and of the positive amide I band of 41° is exceptional small. In the structural model, of all the residues near the retinal, only the backbone C=O group of Glu169 with a relative angle of 47° matches the measured relative angle. Therefore, we assign the bleaching band at 1661 cm^−1^, and the positive band at 1668 cm^–1^ to the C=O backbone group of Glu169, but with distinct dynamics for the loss and regaining of oscillator strengths.

Moreover, the relative angle of the carboxyl C=O group of Glu169 is 28° in the structural model (see Figure 1), which agrees with the measured angle of 29° (2σ: 23–35°) for the bleaching band at 1689 cm^−1^. This corroborates our assignment to the carboxyl C=O group of Glu169. Note, frequencies and orientations of the bleaching signals represent the ground-state structure of *Ca*ChR1.

## 3. Discussion

The observed transient spectra display retinal contributions in line with previous reports, and matching cryo-FTIR difference spectra of *Ca*ChR1 [21]. Moreover, we identified at least four bands reflecting direct ultrafast protein responses. The negative bands at 1689 cm^−1^, and 1661 cm^−1^ are assigned to the carboxylate and backbone C=O groups of Glu169, respectively. These bands appear instantaneously without a positive counterpart, indicating instantaneous loss of oscillator strengths. The assignment of the bleaching band at 1661 cm^−1^ to an amide I vibration is supported by: (i) the frequency position that is typical for amide I absorption of an alpha helix, (ii) the instantaneous occurrence of the bleaching signal demonstrating close vicinity of the vibrational group to the retinal, (iii) the significant signal strength of the bleaching band at 1661 cm^−1^ reflecting strong interaction with the retinal, (iv) the relative angle of the C=O stretching vibration with respect to the electronic tdm, matching the orientation of the C=O backbone group of Glu169. Within tens of picoseconds two positive bands at 1703 cm^–1^, and 1668 cm^−1^ appear, reflecting the rise of a protonated carboxylate group, and the rise of an amide I group, respectively.

From the polarization resolved experiments we found a relative angle of 45°, and 41° for the amide I bands at 1661 cm^−1^, and 1668 cm^−1^, respectively. For rhodopsins, the typical angle between the retinal and the amide I band is around ~70° (see Appendix A, Appendix A). In the structural model, only the backbone C=O group of Glu169 exhibits an angle of 47°, matching the measured values within 2σ error ranges. Thus, we assign both the bleaching band at 1661 cm^−1^, and the product band at 1668 cm^−1^ to the amide I group of Glu169.

Nevertheless, the dynamics of both amide I bands are very different. The bleaching band is observed instantaneously, while the positive band rises with a time constant of 16 ps. The instantaneous loss of oscillator strength of the bleaching band can be explained by a drastic change of the backbone C=O group, i.e., upon protonation and generation of a C-OH group.

The delayed rise of the product band at 1668 cm^−1^ reflects the regeneration of the amide group with a changed force constant. This frequency shift can be explained by structural changes with altered interactions in the photoproduct. Formation of 13-*cis* retinal takes place within 100 fs, followed by cooling and back-reaction that finish within 500 fs and 6 ps, respectively [16,17]. Therefore, retinal related dynamics are negligible on a timescale of tens of picoseconds. However, the rise of the amide I band with 16 ps reflects the delayed cancelling of the induced drastic change of the backbone C=O group of Glu169 in the relaxed photoproduct state. This rise can be explained by deprotonation of the backbone C-OH group of Glu169. The question is, which group serves as a proton donor for ultrafast backbone C=O protonation and which group is the proton acceptor for backbone deprotonation.

Since no deprotonation of the Schiff base is observed, and the fast rise time of 16 ps is not connected to retinal dynamics, we focus on proton donor and acceptor groups of the protein in the vicinity of the retinal. The instantaneous bleaching band at 1690 cm^−1^ is a protein contribution that indicates either deprotonation of a hydrogen-bonded carboxylate group or the loss of oscillator strength of an amide I group [21]. The measured relative angle to the electronic tdm of retinal is between 23° and 35° too low for any amide I group near the retinal, so we can abandon that option. However, the measured angle matches the one of the carboxylate C=O group of Glu169 in the structural model with 28°. Hence, we assign this band to Glu169, which is the only protonated carboxylic acid in the vicinity of the retinal in the ground state [13]. Thus, the instantaneous bleaching band at 1690 cm^−1^ with no direct positive counterpart reflects deprotonation of the COOH group of Glu169. The COOH group of Glu169 is an appropriate proton donor for the backbone amide I group of Glu169, because both groups are close together, and the hydrogen bond between the positively charged Schiff base and the COOH group of Glu169, causing the red-shifted frequency position [22], is perturbed upon photoisomerization.

The bleaching band of the COOH group of Glu169 shows negligible dynamics up to 50 ps, either indicating that this group is not the proton acceptor or that the COOH group of Glu169 exhibits a frequency up-shift to 1703 cm^–1^ in the photoproduct state. The rise of the band at 1703 cm^−1^ with a 16 ps time constant is due to protonation of a COO^−^ group. The proton comes from the backbone CO group of Glu169, reflected by the rise of the Glu169 amide I band at 1668 cm^−1^. After photoisomerization both carboxylate groups of Glu169, and Asp299 are deprotonated. Since both groups are close to the CO backbone group of Glu169, they are valid candidates for proton uptake. Thus, from our data we cannot distinguish whether Glu169 or Asp299 takes up the proton. Cryo-FTIR measurements assigned the band at 1703 cm^−1^ to Asp299, and proposed a direct proton transfer from Glu169 to Asp299 [13].

Combining the polarization resolved data with the structural model of the initial ground state we propose the following reaction mechanism presented in Figure 5. Before photoexcitation, the system is in the electronic ground state (GS) with protonated Glu169, deprotonated Asp299, and the backbone C=O group of Glu169. The backbone C=O group is hydrogen-bonded to the S-H group of Cys173. Upon photoexcitation all-*trans* to 13-*cis* isomerization of the retinal occurs within 100 fs (green arrow in Figure 5). Photoexcitation and retinal isomerization induce a drastic change in electrostatics and steric interactions on the ultrafast timescale. This is accompanied by deprotonation of the COOH group of Glu169 and protonation of its backbone C=O group (green dashed arrow). Since, the SH group of Cys173 is hydrogen-bonded to the backbone C=O group of Glu169, we cannot rule out that the proton transfer could be mediated via the S-H group of Cys173.

At 300 fs the first photoproduct is already established (Figure 5). The retinal structure has changed and Glu169 is deprotonated, reflected by the instantaneous loss of oscillator strength at 1689 cm^−1^ in Figure 3; Figure 4. The instantaneous loss of oscillator strength at 1661 cm^−1^ in Figure 3 and Figure 4 reflects protonation of the backbone CO group of Glu169. This protonation property does not change for about 10 ps, but with a time constant of 16 ps the proton at the backbone CO group of Glu169 is transferred to another carboxylate group. We assign this group to Asp299 [13].

About 30 ps after photoexcitation the photoproduct has changed and the proton from Glu169 is transferred to Asp299, as proposed earlier [13].

In general, it is assumed that protonation of a backbone C=O peptide group is unlikely. However, the *pKa* of a group depends on its local environment: The ultrafast deprotonation of Glu169 alters its charge, and the strong electric field change around the chromophore has an impact on Glu169 [3,6]. Ultrafast structural protein changes were reported for bR by femtosecond X-ray-crystallography [4,5]. Upon photoexcitation, rapid rearrangements near the counterion complex were shown, as well as structural modifications of different helices. However, the reported structural changes observed by femtosecond X-ray-crystallography suffer from too high excitation intensities and multi-photon processes. Thus, ultrafast structural changes in the vicinity of the retinal are considered to be more reliable in a single-photon process compared to changes far away from the retinal. We expect similar rearrangements in *Ca*ChR1 around the chromophore, that could lead to twisting of the Glu169 peptide group, reducing the double-bond character of its C=O [25], and reducing the energy barrier for protonation. Thus, it might be energetically favorable to neutralize the charge at Glu169 by accepting a proton on the backbone CO group. This transient configuration triggers further protonation processes on a timescale of ~20 ps in the photoproduct (see Figure 2, Figure 3 and Figure 4). Thus, structural protein changes seem to be an ongoing process in the first photoproduct already on a timescale of tens of picoseconds.

While the protonation of a protein backbone group is rarely described, there are prior studies [25] proposing that the enolization of such a group is part of the proton transfer pathway in cytochrome c oxidase, based on high-resolution crystal structures [26,27] and supported by DFT calculations [28]. So far, direct spectroscopic evidence of such a process was missing, likely linked to its transient nature. Here we present evidence that in *Ca*ChR1 the transient protonation of a backbone peptide group is part of the initial photoreaction. The spectral main marker of this protonation event, the strong amide I bleaching without supplementing positive band, is also observed in other rhodopsins [7,9,10]. It is not clear if these are due to a similar reaction or caused by another mechanism. Further theoretical studies are required to understand the mechanism, since the cause of the reaction is unknown. Possible contributing factors are the initial electronic field change, charge translocation during the isomerization, and the structural changes due to the isomerization itself.

Since the electrostatic interaction between the retinal, the counterion, and other charged groups is strongly disturbed by the photoexcitation, driving the system far from its equilibrium, structural alterations in the ground state, i.e., by mutations, can disturb the dynamics in unpredictable ways. Thus, mutations around the counterion will not give comparable information on the ultrafast dynamics of wild type *Ca*ChR1. Instead, ultrafast polarization resolved VIS pump—IR probe spectroscopy provides information on dynamics and relative orientations of groups within the protein at sensitive locations next to the chromophore. To further support the proposed reaction model, site-specific isotopic labelling can be used to identify specific groups by spectral shifts without affecting the dynamics.

Applications in optogenetics favor absorption spectra in the red spectral region, since in biological tissues the penetration depth is larger for red light than it is for blue light. Using mutants with charged amino acid side chains in the vicinity of the retinal could change the absorption spectrum to longer wavelengths. The presented study suggests that electric field changes around the retinal could perturb the proton-transfer dynamics, and thus the overall photoreaction quantum yield.

## 4. Materials and Methods

### 4.1. Experimental Methods

#### 4.1.1. Sample Preparation

The sample preparation is described elsewhere [16,17]. In short, the membranous part of *Ca*ChR1 from aa 1-352 with a 10xHis tag at the C-terminus was overexpressed in *Pichia pastoris* and purified via a Ni-NTA column and subsequent gel-filtration. In addition to the described protocol, we exchanged H_2_O with D_2_O in a process of up-concentrating the protein sample in a buffer consisting of 100 mM NaCl, 20 mM Hepes at pH 7.4 and 0.05% DDM.

#### 4.1.2. Spectroscopy

The details of the setup are described elsewhere [17]. In short, the mid-IR probe pulses were generated by a three-stage home-built difference frequency generation set-up at a repetition rate of 1 kHz. The visible pump pulses were generated using a home-built optical parametric amplifier. Pump pulses were centered at 530 nm, had an energy of 400 nJ to 600 nJ and a focus diameter of 200 µm. The polarization was switched each time after a scan of all delay time points. The sample was placed between two CaF_2_ windows separated by a 50 µm teflon spacer. During the experiment the sample was rapidly moved by a home-built Lissajous scanner to guarantee a fresh sample spot for every pump pulse. The spectral resolution was 2 cm^–1^ for the dataset between 1530 cm^–1^ and 1670 cm^–1^ and 4 cm^–1^ for the dataset covering the carboxyl stretching region around 1700 cm^–1^. The time-resolution of ~350 fs was given by the measured system response in a Ge-wafer.

Using photoselection and polarization resolved probing we determine the relative angle Φ between the initial electronic transition dipole moment (tdm) of the S_0_ → S_1_ transition and the vibrational tdm of vibrational transitions. Since the electronic tdm is fixed within the retinal molecule and the vibrational tdm is fixed to the vibrating group, the relative angle provides structural information of the involved states and orientational changes [19]. The relative angle Φ is related to the dichroic ratio d, $d=A_{∥}/A_{⊥}$, via the absorption strengths for parallel ($A_{∥}$) and perpendicular $(A_{⊥}$) polarizations.
(1)$$\phi \left(d\right)=\;\arccos\sqrt{\left(\frac{2d-1}{d+2}\right)}$$

The measured relative angles Φ of vibrational bands can be related to structural features with a given structural model.

### 4.2. Theoretical Methods

#### 4.2.1. Structural Modelling

A homology model of *Ca*ChR1 was generated based on the crystal structures of related channelrhodopsins. This model enabled the computation of excitation energies and transition dipole moments. Three available ChR crystal structures that share a high sequence identity with *Ca*ChR1, namely C1C2 [29] (PDB ID: 3UG9, 53% identity), *Cr*ChR2 [30] (PDB ID: 6EID, 57% identity) and Chrimson [31] (PDB ID: 5ZIH, 49% identity) were combined to generate the initial model. All four protein sequences were aligned with Clustal Omega [32] (version 1.2.4, University College Dublin Conway Institute, Dublin, Ireland), and MODELLER [33] (version 9.20, Laboratory of Andrej Sali, University of California, San Franciso, San Francisco, CA, USA) and was used to generate a homology model. In the model, standard protonation states were used for all residues, except for Cys174 and Asp202, which were deprotonated and protonated, respectively in the ground state. The generated model was optimized with hybrid quantum mechanics/molecular mechanics (QM/MM) using the ChemShell [34,35] software package (version 3.7.0, STFC Daresbury Laboratory, Daresbury, UK). The model was optimized until the default ChemShell convergence criteria for changes in energy, gradients and step size were met. The QM region consisted of the retinal chromophore and the Lys303 side chain. The QM method during the optimization was the CAM-B3LYP method [36]. For the MM part, the CHARMM36 protein force field [37,38] and the TIP3P water model [39] were employed throughout all our computations. The Amber [40] QM/MM interface [41] was used to generate final input files from the optimized structure. The excitation energies and associated tdms of the five lowest excited states were then computed using the second-order approximate coupled-cluster singles and doubles model CC2 [42] with the resolution-of-identity approximation (RI). The basis set was cc-pVDZ with the corresponding auxiliary basis set [43]. Turbomole (version 7.3, COSMOlogic, Leverkusen, Germany) [44,45] was used for RI-CC2 calculations.

#### 4.2.2. Estimation of Vibrational tdm Orientation

An accurate calculation of all vibrational modes in a large protein as *Ca*ChR1 is unattainable. Hence, we use a simpler approach and just estimate the direction of vibrational tdms from the structure via simple heuristics: For the carboxylic acid C=O stretching vibrations we assume the tdm is orientated along the C=O bond direction. The C=O vibration of the protein backbone (amide I) was shown to couple to the C-N stretching and N-H-bending mode of the peptide group, resulting in a rotation of the vibrational tdm by ~10° (20° in H_2_O) in the NCO plane towards the nitrogen with respect to the C=O direction (Figure 1) [20]. Note, that this approach neglects coupling between neighboring amide I vibration.

## 5. Conclusions

By using ultrafast polarization-resolved mid-IR spectroscopy, we are able to dissect parts of the initial photoreaction of *Ca*ChR1. Besides the isomerization of the retinal, we observe ultrafast proton transfer from Glu169 to its backbone CO group. This structural change is an immediate response to the isomerization process. The transient protonation of the Glu169 backbone establishes an instable configuration, resulting in a proton transfer from the backbone to Asp299. This step occurs on a timescale of tens of picoseconds and is not related to retinal dynamics anymore, but to ongoing dynamics of the altered protein structure. These structural changes are expected to be connected or to initiate changes on the nanosecond to microsecond scale. We assume these protein dynamics to be the first steps in a cascade of wider protein changes resulting in channel conductivity. Thus, these processes have to be considered for developing efficient mutants for optogenetics with more red-shifted absorption spectra.

This is, to our knowledge, the first report of ultrafast proton-transfer reactions in rhodopsins; it is also the first spectroscopic evidence of backbone groups being an active part of a proton-transfer pathway. Moreover, the transient protonation is responsible for strong amide I changes, which are therefore not caused by large structural rearrangements. Hence the presented results show, that the correlation of amide I changes to structural rearrangements has to be questioned if a significant loss of amide I oscillator strength is observed. Since similar amide I signatures are also observed in other rhodopsins, especially in *Cr*ChR2, similar ultrafast protonation events may also participate in their photoreaction.

## Figures and Tables

**Figure 1 molecules-25-00848-f001:**
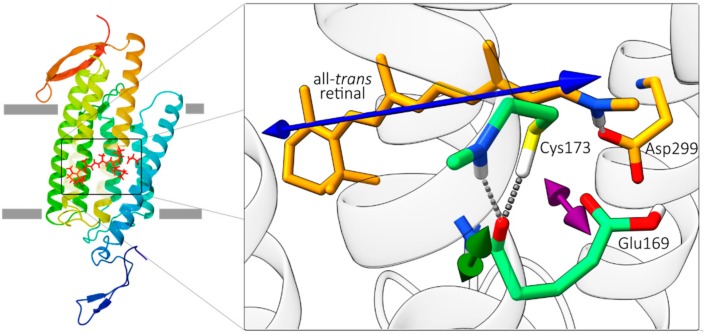
The calculated ground-state structure of *Chlamydomonas augustae* (*Ca*ChR1). For clarity, the figure shows only selected amino acid residues. The blue arrow displays the direction of the calculated electronic transition dipole moment (tdm) of the S_0_ to S_1_ transition and the green and the purple arrows show the vibrational tdm direction of the amide I mode of Glu169 and of the carboxyl C=O group of Glu169.

**Figure 2 molecules-25-00848-f002:**
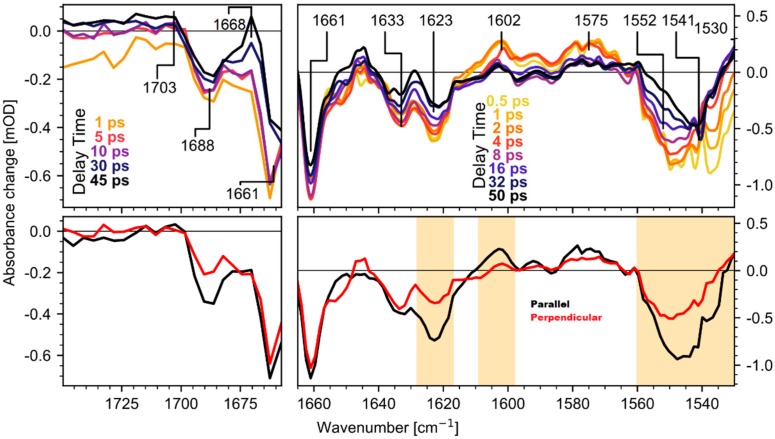
Transient absorption spectra of *Ca*CR1 after excitation at 530 nm in the range of 1730 cm^−1^ to 1656 cm^−1^ (left panels) with a spectral resolution of 4 cm^−1^, and 1665 cm^−1^ to 1530 cm^−1^ (right panels) with a spectral resolution of 2 cm^−1^. Top row: Isotropic absorption changes for selected delay times for both spectral regions. Bottom row: From 3 ps to 8 ps averaged absorption changes for parallel (black lines) and perpendicular (red lines) polarizations of pump and probe. Orange shaded regions indicate spectral regions with dominating retinal contributions.

**Figure 3 molecules-25-00848-f003:**
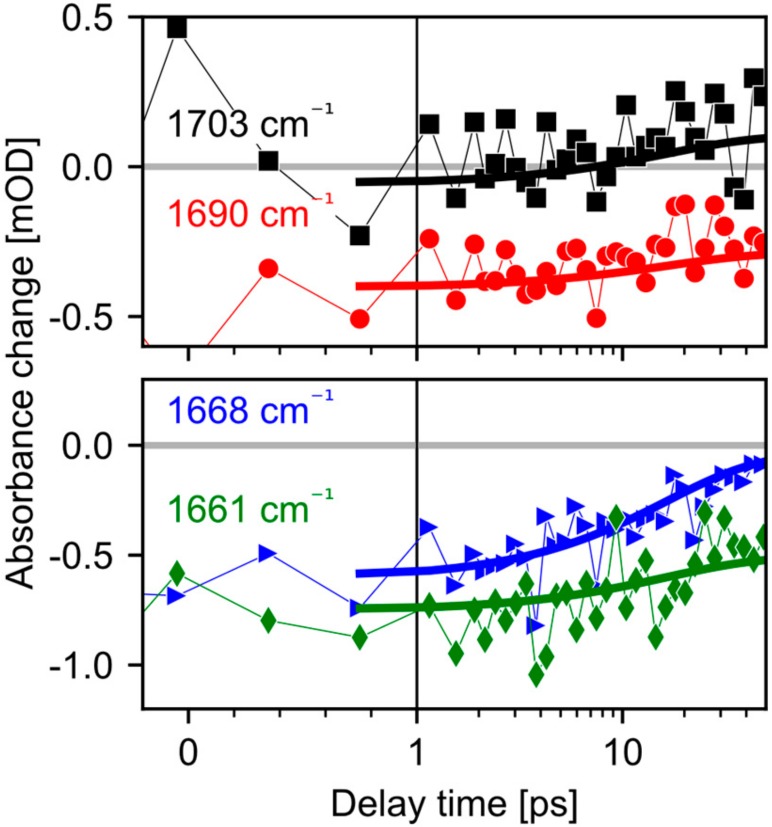
Transients for parallel polarization of pump and probe comparing the dynamics of the protein bands in the amide I and carboxyl region. Symbols represent raw data while the thick lines display the fitted model. The slight decay of the amide I bleaching band at 1661 cm^−1^ (green diamonds) is likely caused by the rise of the adjacent 1668 cm^−1^ band (blue triangles). The process is accompanied by the rise of the 1703 cm^−1^ band (black squares). On the same timescale, the negative band near 1690 cm^−1^ (red circles) remains nearly unchanged.

**Figure 4 molecules-25-00848-f004:**
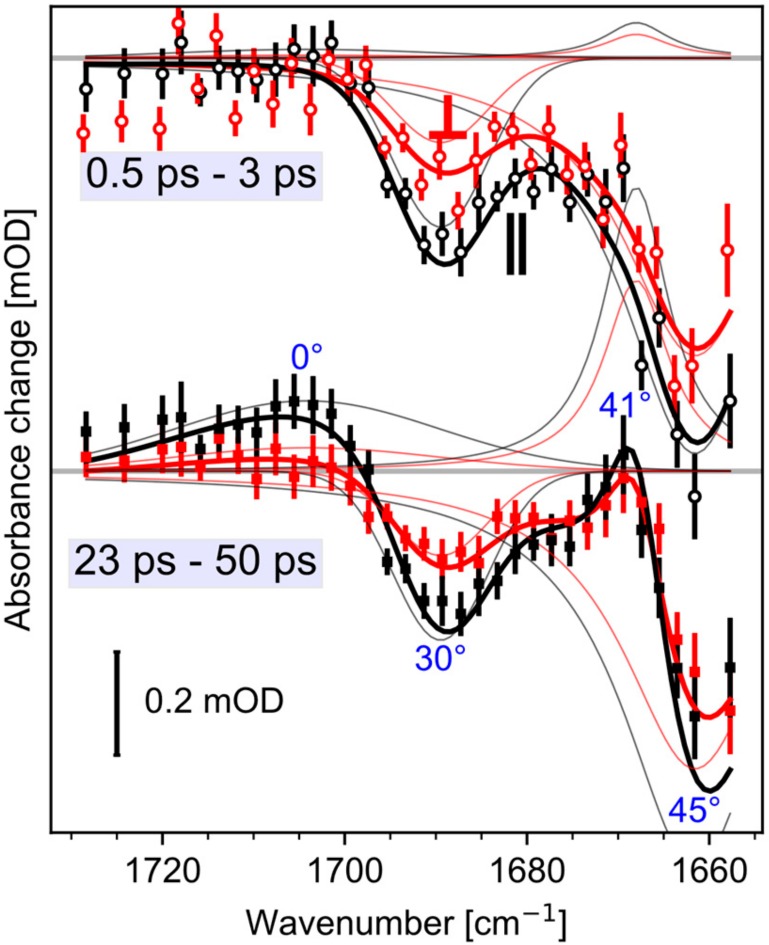
Peak decomposition of the polarization resolved transient spectra in amide I and carboxyl stretching region. Shown are the averaged measured and fitted spectra for early (top) and late (bottom) delay times. Absorption changes for parallel polarization (black lines), and perpendicular polarization (red lines). The thick black and red lines show the fitted curves and the markers show the measured spectra. Thin black and red lines show the decomposed peaks for parallel and perpendicular polarizations, respectively. The dichroic ratio can be obtained by the ratio of parallel and perpendicular polarized absorption strengths of the decomposed peaks. The resulting angles calculated by Equation (1) are given in blue for both time windows.

**Figure 5 molecules-25-00848-f005:**
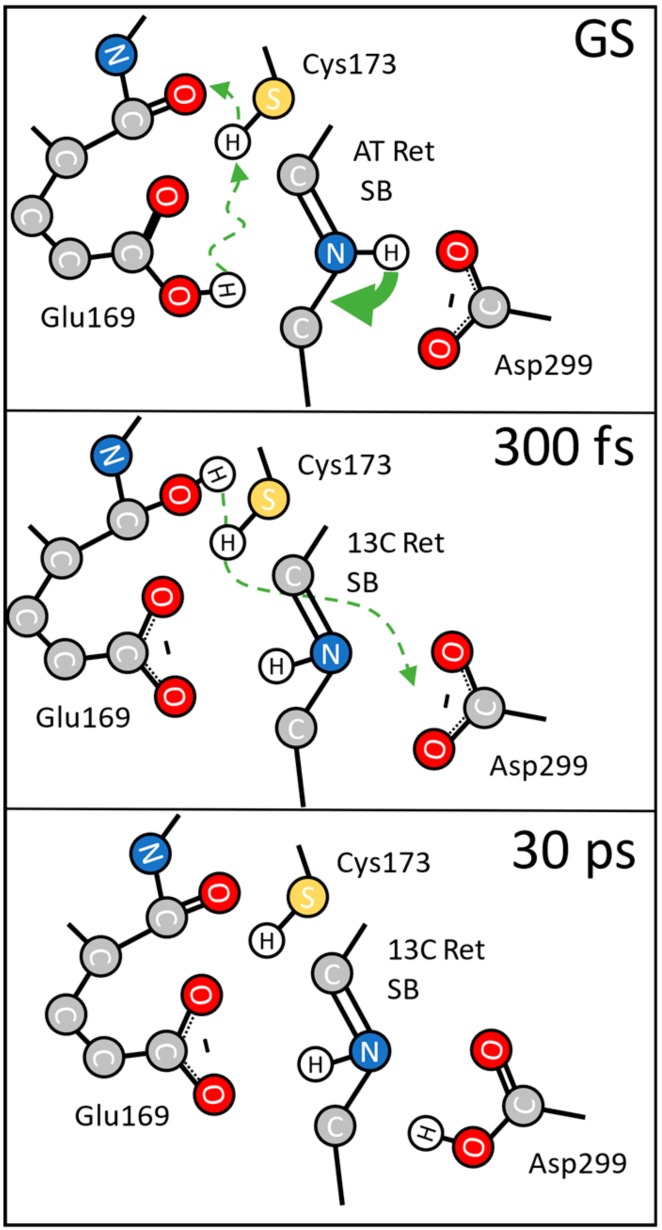
Proposed reaction scheme for the protein response after photoexcitation. In the ground state (GS) Glu169 is protonated. Upon excitation the retinal isomerizes within 100 fs (green arrow) and induces deprotonation of Glu169 (green dashed arrow); the proton is transferred to the nearby backbone CO-group of Glu169 itself; the SH group of Cys173 may be involved. At 300 fs the photoproduct is formed with the Schiff base NH group pointing in a different direction; we have deprotonated carboxylate groups of Glu169, and Asp299; the backbone CO group of Glu169 is protonated. With a decay time of 16 ps the proton at the backbone CO group of Glu169 is transferred to Asp299. At 30 ps Glu169 is deprotonated, while Asp299 is protonated.

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
