# Peer review of "Ultrafast Backbone Protonation in Channelrhodopsin-1 Captured by Polarization Resolved Fs Vis-pump—IR-Probe Spectroscopy and Computational Methods"

_molecules, 2020, doi:10.3390/molecules25040848_

Round 1

Reviewer 1 Report

Comment to the authors

In this manuscript (Manuscript ID: molecules-702160), Stensitzki et al. studied the dynamic structural change of channelrhodopsin-1 from Chlamydomonas augustae (CaChR1) upon the photo-excitation by the polarization resolved ultrafast mid-infrared spectroscopy, and the structure of CaChR was modelled by quantum mechanics/molecular mechanics (QM/MM) calculation to assign the experimentally observed vibrational modes to each residue. Their ultrafast mid-IR spectroscopy was highly precise and they succeeded in observing IR absorption change whose intensity was less than 1 mOD in 1530-1750 cm-1 region. Based on the result, the authors suggested ultrafast transient proton transfers occur between carboxylic group and peptide backbone in the counterion complex. However, despite of the model including untypical protonation change of peptide backbone, the assignment was mainly carried out only by comparing the relative angle obtained by the experiment and modelled structure. Also, the reliability of the modelled structure was not discussed in the manuscript. To conclude the untypical model, the authors should more carefully interpret their results by adding more experimental supporting data (see below), and this reviewer considers this paper is mandatory to be comprehensively revised for the publication at several points shown below

Major points

#The biggest concern of this reviewer is the assignment of the 1661(-) cm-1 peak for the deprotonation of backbone of Glu169. This conclusion was derived from the comparison between the relative angles observed in the experiment and in the modelled structure (Figure S1). However, the authors showed only selected 13 out of 352 residues in the figure and all of them are located only in 3rd and 7th helices. Recent time-resolved serial femtosecond crystallography (TR-SFX) of BR showed rapid structural change of 6th helix occurs in the picosecond time region due to the steric repulsion between the C20-methyl group of retinal chromophore and Trp (Nogly et al. Science (2018)). Therefore, the authors need to more carefully discuss the residues in other helices, not only 3rd and 7th helices, especially for 6th helix. Furthermore, in order to assign the 1661(-)-cm-1 band to the transient proton transfer from Glu169 to its backbone carbonyl, an experiment with a mutant of Glu169 (CaChR1 E169Q) in which proton transfer is inhibited would give the most direct evidence. To support the conclusion in which transient proton transfer between carboxylic group and backbone carbonyl is involved, further experiment with mutant protein is needed.

#The reliability of the structure modelled by QM/MM calculation should be more clearly described. Although the authors wrote that “Details of the procedure are given in the supplementary information section S2.” (Page 3, lines 92-93), this reviewer cannot find such description in the supplementary information section S2. It should be clearly stated did the structure converge with how long simulation time of QM/MM calculation.

#A deprotonation of Glu90 in ChR2 in 2nd helix faster than 1 ns was reported by time-resolved FTIR spectroscopy (Kuhne et al. PNAS (2019)). Since the identical glutamic acid is conversed in CaChR1, the authors should discuss the possibility that the acidic residues in other helices, especially 2nd helix of CaChR1 has more than 3 acidic residues including a glutamic acid homologous to ChR2 Glu90, involved in the rapid proton transfer.

Minor points

#Figure 2

The time evolution of the intensity of 1661(-)-cm-1 peak is different between left and right figures. While ca 25% reduction of the peak intensity was observed and the peak shape was not drastically changed at 32-50 ps in right figure, the peak becomes a shoulder at 31-50 ps in the left figure. Why do they apparently differ from each other?

#Some figures are placed distant from related text, e. g. Figures 1 and 2 are placed on page 2 and 3, but they are first mentioned on page 3 and 4 respectively. It is better to arrange the figures near the relevant text for readability.

#Page 4, line 133. “This indicates formation of a new positive band at 1668 cm-1.”

It is better to add the corresponding tag in Figure 2.

#Figure 4

Decomposed peaks at 1660(-) cm-1 are lacked for the 0.5-3 ps data (top).

#The abbreviation of channelrhodopsin-1 from Chlamydomonas augustae is not fixed such as CaChR1, CaChR1, CaChr1. The same abbreviation should be used throughout the manuscript. Also, CrChr2 and SR2 (Page 2, lines 44-45) are not common abbreviations, and it is better to use CrChR2 and SRII .

#Page 9, Materials and Methods

The description about protein sample is not enough. Was the protein put into detergents or lipids? Also, chemical composition of solvent is mandatory to be described for future experiment to reproduce their restuls.

#Some of the numbering of amino acid residues might be needed to be corrected, e. g. “Glu196 (Page 8, line 253)”, “Cys172 and Asp202 (Page 10, line 336)”, “Lys303 (Page 10, line 340)”, and so on.

Author Response

First of all we would like to thank reviewer 1 for effort spent to identify unclear points to optimize the final manuscript. Here, we explain in detail our changes performed in the manuscript and explained points where and why we could not follow reviewer's suggestions in their entirety.

However, despite of the model including untypical protonation change of peptide backbone, the assignment was mainly carried out only by comparing the relative angle obtained by the experiment and modelled structure.

The assignment was carried out by (i) the frequency position that is typical for amid I absorption of an alpha-helix, (ii) the instantaneous occurrence of the bleaching signal demonstrating close vicinity of the vibrational group to the retinal, (iii) the significant signal strength of the bleaching band at 1661 cm-1 reflecting strong interaction with the retinal, (iv) the relative angle of the C=O stretching vibration with respect to the electronic tdm.

We added to the discussion (lines 191-196): “The assignment of the bleaching band at 1661 cm-1 to an amid I vibration is supported by (i) the frequency position that is typical for amid I absorption of an alpha-helix, (ii) the instantaneous occurrence of the bleaching signal demonstrating close vicinity of the vibrational group to the retinal, (iii) the significant signal strength of the bleaching band at 1661 cm-1 reflecting strong interaction with the retinal, (iv) the relative angle of the C=O stretching vibration with respect to the electronic tdm, matching the orientation of the C=O backbone group of Glu169.”.

2. Also, the reliability of the modelled structure was not discussed in the manuscript. To conclude the untypical model, the authors should more carefully interpret their results by adding more experimental supporting data (see below), and this reviewer considers this paper is mandatory to be comprehensively revised for the publication at several points shown below:

We added more information to the reliability of the modelled ground state structure (see below).

Major points:

3. #The biggest concern of this reviewer is the assignment of the 1661(-) cm-1 peak for the deprotonation of backbone of Glu169. This conclusion was derived from the comparison between the relative angles observed in the experiment and in the modelled structure (Figure S1). However, the authors showed only selected 13 out of 352 residues in the figure and all of them are located only in 3rd and 7th helices. Recent time-resolved serial femtosecond crystallography (TR-SFX) of BR showed rapid structural change of 6th helix occurs in the picosecond time region due to the steric repulsion between the C20-methyl group of retinal chromophore and Trp (Nogly et al. Science (2018)). Therefore, the authors need to more carefully discuss the residues in other helices, not only 3rd and 7th helices, especially for 6th helix.

Firstly, the assignment of the band at 1661 cm-1 to Glu169 has different reasons. We want to thank the reviewer to point out that this was not clear from our manuscript. Thus, we added the lines 191-196 “The assignment…Glu169.

Furthermore, the reviewer pointed out that the Papers of Nogly et al. and Kovacs et al. Nature Com 2019 showed rapid structural changes in bR in different helices. In the paper of Nogly et al (Figure S5 supplementary information) there are indeed hints of structural changes in helices B and G, but these features are much weaker than those around the retinal chromophore. Nogly et al stated in this manuscript:

Protein quake: The collective motions from the counterion region extend up to 12 Å away from the SB nitrogen within 600 fs (fig. S5) yet become dispersed on longer time scales. This dampening is evidenced by weaker paired difference densities associated with Tyr57 at Dt = 759 to 1025 fs, as well as the motions of Thr47 and Asp212, which are no longer visible at Dt = 10 ps. Changes are further apparent when comparing the refined structural intermediates at Dt = 49 to 406 fs, 457 to 646 fs, and 10 ps (Movie 2). As such, these motions propagate away from the active site at 2 nm/ps, which is noticeably faster than the speed of sound in water (~1.5 nm/ps). These observations are consistent with the propagation of motions in the bacterial photoreaction center (37) and myoglobin (16, 38, 39) and the theory of protein quakes that suggests how excess energy in proteins can be dissipated in earthquake-like motions of collective structural deformations (40).”

Since the bleaching band at 1661 cm-1 is observed instantaneously (within our system response of ≤ 350 fs), the reported motions should be closer to the active site than 7Å. Furthermore, it is under debate whether a significant part of the protein quake is induced by the strong multi-photon processes in these experiments.

Thus, we added in lines 277-284 “Upon photoexcitation, rapid rearrangements near the counterion complex were shown, as well as structural modifications of different helices. However, the reported structural changes observed by femtosecond x-ray-crystallography suffer from too high excitation intensities and multi-photon processes. Thus, ultrafast structural changes in the vicinity of the retinal are considered to be more reliable in a single-photon process compared to changes far away from the retinal. We expect similar rearrangements in CaChR1 around the chromophore, that could lead to twisting of the Glu169 peptide group, reducing the double bond character of its C=O, [25] and reducing the energy barrier for protonation.

4. Furthermore, in order to assign the 1661(-)-cm-1 band to the transient proton transfer from Glu169 to its backbone carbonyl, an experiment with a mutant of Glu169 (CaChR1 E169Q) in which proton transfer is inhibited would give the most direct evidence. To support the conclusion in which transient proton transfer between carboxylic group and backbone carbonyl is involved, further experiment with mutant protein is needed.

We agree with the reviewer that a change of the geometry at Glu169 or other changes at the counterion complex will impact the ultrafast interaction of retinal, amino acid groups and water. But in which way it alters the ultrafast dynamics is not predictable. The proposed change of an Glu group to Gln would change the sensitive interaction pattern of the counterion complex with the Schiff base and water molecules, resulting in altered hydrogen bonding pattern. It is definitely interesting to investigate ultrafast dynamics of the E169Q mutant, but the local geometry and its resulting interaction pattern and dynamics are not comparable. To strengthen this point in our manuscript we added on lines 300-307:

Since the electrostatic interaction between the retinal, the counterion, and other charged groups is strongly disturbed by the photoexcitation, driving the system far from its equilibrium, structural alterations in the ground state, i.e. by mutants, can disturb the dynamics in unpredictable ways. Thus, mutations around the counterion will not give comparable information on the ultrafast dynamics of wild type CaChR1. Instead, ultrafast polarization resolved VIS pump – IR probe spectroscopy provides information on dynamics and relative orientations of groups within the protein at sensitive locations next to the chromophore. To further support the proposed reaction model site specific isotopic labelling can be used to identify specific groups by spectral shifts without affecting the dynamics.

Moreover, the C=O group of the corresponding group of E90 in the homology model exhibits an angle close to 90° to the electronic tdm, not matching with our measured angle of the bleaching band at 1661 cm-1. Furthermore, we added the pdb-file (Homology model CaChr1.pdb) of the calculated homology model to the supporting information.

5. #The reliability of the structure modelled by QM/MM calculation should be more clearly described. Although the authors wrote that “Details of the procedure are given in the supplementary information section S2.” (Page 3, lines 92-93), this reviewer cannot find such description in the supplementary information section S2. It should be clearly stated did the structure converge with how long simulation time of QM/MM calculation.

We apologize for the wrong reference. All the details about the methodology are found in the Methods section lines 362-384.

We have done a geometry optimization of the protein model using QM/MM and not a molecular dynamics simulation. Therefore there is no information about the time. Our geometry optimization has fully converged, all five convergence criteria were fullfilled (1) Energy change, 2) Max gradient, 3) RMS gradient, 4) Max step size, 5) RMS steps size). This is now explicitly mentioned in the manuscript lines 73-77:

” 2.1 Computational results

Since no structure of CaChR1 is available yet, we apply homology modelling and QM/MM geometry optimization to obtain a ground-state structural model of CaChR1. The part of the structural model around the retinal chromophore is depicted in Figure 1. ”

6. #A deprotonation of Glu90 in ChR2 in 2nd helix faster than 1 ns was reported by time-resolved FTIR spectroscopy (Kuhne et al. PNAS (2019)). Since the identical glutamic acid is conversed in CaChR1, the authors should discuss the possibility that the acidic residues in other helices, especially 2nd helix of CaChR1 has more than 3 acidic residues including a glutamic acid homologous to ChR2 Glu90, involved in the rapid proton transfer.

The mentioned manuscript by Kuhne et al. PNAS 2019 exhibits no time-resolved experiments below the ms time-scale. But in this manuscript another paper of Kuhne et al (2015) is mentioned:

“Time-resolved FTIR spectroscopy was originally established as a powerful approach for the determination of the molecular reaction mechanism of BR (19). Accordingly, the dark-adapted ChR2 photocycle was recorded between 50 ns and 140 s after exposure to a light pulse by step-scan and rapid-scan FTIR. These measurements revealed an ultrafast all-trans to 13-cis isomerization and subsequent deprotonation of the RSBH+ in parallel with protonation of the counter-ion residues E123 and D253 (18).”.

In the other manuscript transient changes from ns to seconds were reported. In our manuscript we investigate much faster processes at least one order of magnitude faster. In order to stress that further protein changes are expected on longer time-scales we introduced on lines 320-321: 

“These structural changes are expected to be connected or initiate changes on the nanosecond to microsecond time-scale.”

Minor points

7. #Figure 2

The time evolution of the intensity of 1661(-)-cm-1 peak is different between left and right figures. While ca 25% reduction of the peak intensity was observed and the peak shape was not drastically changed at 32-50 ps in right figure, the peak becomes a shoulder at 31-50 ps in the left figure. Why do they apparently differ from each other?

The dynamics of the 1661 (-) cm-1 peak is not noticeable different in the left and right figures. The shoulder at 30-45 ps in the left figure is at higher wavenumbers (at 1668 cm-1) and is also indicated in the right figure (highest frequency at 1665 cm-1) at 50 ps where the absorption crosses the zero line. However, the different spectral resolution (2 cm-1 and 4 cm-1) could be a reason for this impression.

8. #Some figures are placed distant from related text, e. g. Figures 1 and 2 are placed on page 2 and 3, but they are first mentioned on page 3 and 4 respectively. It is better to arrange the figures near the relevant text for readability.

We arranged all Figures next to the related text.

9. #Page 4, line 133. “This indicates formation of a new positive band at 1668 cm-1.”

It is better to add the corresponding tag in Figure 2.

We added a tag to Figure 2.

10. #Figure 4

Decomposed peaks at 1660(-) cm-1 are lacked for the 0.5-3 ps data (top).

The decomposed peaks at 1660 (-) cm-1 are not lacking for the 0.5-3 ps data (top). The lines have the same thickness as the other lines for decomposed peaks. Increasing the thickness of all lines for decomposed peaks makes the figure too busy. We added in the figure caption: “Thin black and red lines show the decomposed peaks for parallel and perpendicular polarizations, respectively. The dichroic ratio can be obtained by the ratio of parallel and perpendicular polarized absorption strengths of the decomposed peaks. The resulting angles are given in blue for both time windows.

11. #The abbreviation of channelrhodopsin-1 from Chlamydomonas augustae is not fixed such as CaChR1, CaChR1, CaChr1. The same abbreviation should be used throughout the manuscript. Also, CrChr2 and SR2 (Page 2, lines 44-45) are not common abbreviations, and it is better to use CrChR2 and SRII .

We changed the notation in the manuscript to CaChR1, CrChR2 and SRII.

12. #Page 9, Materials and Methods

The description about protein sample is not enough. Was the protein put into detergents or lipids? Also, chemical composition of solvent is mandatory to be described for future experiment to reproduce their restuls.

We added to the materials and methods lines 338-340:

“In addition to the described protocol, we exchanged H2O with D2O in a process of up-concentrating the protein sample in a buffer consisting of 100 mM NaCl, 20mM Hepes at pH 7.4 and 0.05% DDM.”

13. #Some of the numbering of amino acid residues might be needed to be corrected, e. g. “Glu196 (Page 8, line 253)”, “Cys172 and Asp202 (Page 10, line 336)”, “Lys303 (Page 10, line 340)”, and so on.

We corrected the typos.

After these revisions we feel that we have addressed all points raised by reviewer 1 in detail.

Reviewer 2 Report

The manuscript reports characterisation of the retinal chromophore isomerisation and the channelchhodopsin-1 channel opening. This study was performed by combining of homology modelling/MD simulations with polarisation resolved femtosecond mid-IR spectroscopy to characterize photo-excited reaction dynamics of channelrhodopsin-1 at a high time resolution. the direct transient information from the spectroscopy data to study the relative orientation changes of the vibrational modes was accessed.

Principle remarks:

Modelling parts (both in Methods and in Results) are described very briefly such opening to different questions.

Which is similarity/identity of the target protein and used templates?  It’s no clear, did the stability of the homology model was investigated by molecular dynamic (MD) simulation or not? Which is quality of the model before the QM/MM simulations? Did the protein/retinal interaction energy (free binding energy) was estimated? Such estimation may help in calculation the interaction energy for each residue, an essential data to explain “contributions and dynamics of individual amino acid groups” (lines 66-67).

The absence of the MD-based characterisation of the protein flexibility limits seriously the interpretation of data in terms of “structural changes”, “protein dynamics”, “direct contact”, “hydrogen bond”, or “energetically favourable”, everywhere used by the authors. Such interpretation requires the excellent level of refined 3D model supported by statistically valid data (through MD simulations), which in turn can be used for the required/related calculations (structural changes, protein dynamics, direct contacts, hydrogen bonds or free binding energy).

In discussions, a role of the S-H group of Cys172 is not clear. Why its involving in the reaction was suggested? The figure 5 will be a more comprehensive if the schemes will have completed by the 3D structures (for example, snapshots from simulations).

Experimental part: The authors previously have reported the vibrational dynamics of photoisomerization in channelrhodopsin-1 that was limited to C-C stretching region (Ref. 16 and 17). In the present manuscript the experiments were conducted similarly, for the region of C-X stretching modes to observe the relative orientation changes. These measurements permitted to postulate ultrafast proton transfer reaction in rhodopsin.

Other remarks:

Line 89: “Theoretical results” should be Computational results; idem on line 328 Line 84: Figure 2. The delta symbol of the absorption change unit in the vertical axis label is omitted. line 92: It was mentioned that “Details of the procedure are given in the supplementary information sectionS2” (line 92). In reality, S2 reports “Comparison of the measured angles with angles in the homology model”. Lines 110-119 (Experimental results) contain information related to the method description. Line 160: Figure 3 should be Figure 4 “Our model” or “our structural model” should be everywhere “the model” or “the 3D model” or “homology model”. “Our” is not the appropriate terms in this context. Similarly, “we have a deprotonated carboxylate groups” (line 214) and other very personalised expression, like “our polarization”. What do you mean by “protein substructure”? (line 261) Extensive revision of English and of the used terms  is required.

Author Response

We thank reviewer 2 for the constructive criticism and time spent to improve our manuscript. We changed several points and corrected misunderstandings raised by our manuscript:

 The manuscript reports characterisation of the retinal chromophore isomerisation and the channelchhodopsin-1 channel opening. This study was performed by combining of homology modelling/MD simulations with polarisation resolved femtosecond mid-IR spectroscopy to characterize photo-excited reaction dynamics of channelrhodopsin-1 at a high time resolution. the direct transient information from the spectroscopy data to study the relative orientation changes of the vibrational modes was accessed.

Principle remarks:

Modelling parts (both in Methods and in Results) are described very briefly such opening to different questions.

1. Which is similarity/identity of the target protein and used templates?

This information is now included in the manuscript (lines 368-369). 

2. It’s no clear, did the stability of the homology model was investigated by molecular dynamic (MD) simulation or not?

No, we have done a geometry optimization in the QM/MM framework instead of molecular dynamics. We changed lines 73-77:

“2.1 Computational results

Since no structure of CaChR1 is available yet, we apply homology modelling and QM/MM geometry optimization to obtain a ground-state structural model of CaChR1. The part of the structural model around the retinal chromophore is depicted in Figure 1.”

3. Which is quality of the model before the QM/MM simulations? Did the protein/retinal interaction energy (free binding energy) was estimated?

No, we have not calculated the free binding energy because retinal is covalently bound to the protein and is located inside the binding pocket. 

4. Such estimation may help in calculation the interaction energy for each residue, an essential data to explain “contributions and dynamics of individual amino acid groups” (lines 66-67).

We believe this is a misunderstanding. This work is focused on a proton transfer upon photoisomerization, which is a non-equilibrium process. We don’t understand why the interaction energy for each residue is required? 

5. The absence of the MD-based characterisation of the protein flexibility limits seriously the interpretation of data in terms of “structural changes”, “protein dynamics”, “direct contact”, “hydrogen bond”, or “energetically favourable”, everywhere used by the authors.

We would like to emphasize that the terms mentioned by the reviewer are experimental findings. The experiment tracks a light-induced process and the associated conformational changes. In contrast, the simulation in this work are describing the initial starting point in the ground state. Hence we cannot compare the simulations directly to the time-resolved experiment because for that we would have to simulate the photoisomerization and the consequent dynamics of the protein. 

6. Such interpretation requires the excellent level of refined 3D model supported by statistically valid data (through MD simulations), which in turn can be used for the required/related calculations (structural changes, protein dynamics, direct contacts, hydrogen bonds or free binding energy).

In principle we agree, but in this specific case we have used a homology model based on three structures with a similarity of > 60%. 

7. 20.) In discussions, a role of the S-H group of Cys172 is not clear. Why its involving in the reaction was suggested?

We added the lines 250-253:

“Since, the SH group of Cys173 is hydrogen bonded to the backbone C=O group of Glu169, we cannot rule out that the proton transfer could be mediated via the S-H group of Cys173.”

8. 21.) The figure 5 will be a more comprehensive if the schemes will have completed by the 3D structures (for example, snapshots from simulations).

We have not performed MD simulations on the photoproduct. These will be extremely difficult if one has to take into account the instantaneous electric field change in the electronic excited state and the change of electrostatics during photoisomerization. Thus, we started from an optimized ground state structure before excitation and compared this structure with our vibrational bleaching signals. Positive signals in the IR-data indicate photoproduct signatures. These were discussed on the basis of our femtosecond polarization resolved VIS-pump IR-probe data.

Experimental part: The authors previously have reported the vibrational dynamics of photoisomerization in channelrhodopsin-1 that was limited to C-C stretching region (Ref. 16 and 17). In the present manuscript the experiments were conducted similarly, for the region of C-X stretching modes to observe the relative orientation changes. These measurements permitted to postulate ultrafast proton transfer reaction in rhodopsin.

Other remarks:

22.) 1. Line 89: “Theoretical results” should be Computational results; idem on line 328

Done.

23.) 2. Line 84: Figure 2. The delta symbol of the absorption change unit in the vertical axis label is omitted.

Done.

24.) 3. line 92: It was mentioned that “Details of the procedure are given in the supplementary information sectionS2” (line 92). In reality, S2 reports “Comparison of the measured angles with angles in the homology model”.

A detailed discussion is now presented in the supporting information section S1. We changed the text of the manuscript accordingly.

25.) 4. Lines 110-119 (Experimental results) contain information related to the method description.

We moved this part to materials and methods lines 352-361.

26.) 5. Line 160: Figure 3 should be Figure 4

Done.

27.) 6. “Our model” or “our structural model” should be everywhere “the model” or “the 3D model” or “homology model”. “Our” is not the appropriate terms in this context. Similarly, “we have a deprotonated carboxylate groups” (line 214) and other very personalised expression, like “our polarization”.

We changed all personalized expressions in the text.

28.) 7. What do you mean by “protein substructure”? (line 261)

We changed the phrase on lines 267-269 to

“This protonation property does not change for about 10 ps, but with a time constant of 16 ps the proton at the backbone CO group of Glu169 is transferred to another carboxylate group.”

29.) 8. Extensive revision of English and of terms using is required.

We improved the abbreviations and some text passages.

After these revisions we feel that we have addressed all points raised by reviewer 2 in detail.

Reviewer 3 Report

The whole manuscript is well written. The topic is new and attractive. Authors show the ultrafast proton transfer reactions in rhodopsins. They claim that the transient protonation is responsible for strong amide I changes, which are  not caused by large structural rearrangements - the correlation of amide I changes to structural rearrangements is not always valid. More efford should be given to explanation under which conditions the  amide I changes are not caused by  large structural rearrangements i.e. why and under which conditions  the correlation of amide I changes to structural rearrangements is not valid.

Author Response

We want to thank reviewer 3 for assessing our manuscript, for the positive feedback, and for the help to stress one of our major findings:

The whole manuscript is well written. The topic is new and attractive. Authors show the ultrafast proton transfer reactions in rhodopsins. They claim that the transient protonation is responsible for strong amide I changes, which are  not caused by large structural rearrangements - the correlation of amide I changes to structural rearrangements is not always valid.

More efford should be given to explanation under which conditions the  amide I changes are not caused by  large structural rearrangements i.e. why and under which conditions  the correlation of amide I changes to structural rearrangements is not valid.

To clarify this point we modified the lines 329-331: “Hence the presented results show, that the correlation of amide I changes to structural rearrangements has to be questioned if a significant loss of amid I oscillator strength is observed.”.

After these revisions we feel that we have addressed the point raised by reviewer 3.

Round 2

Reviewer 1 Report

The authors appropriately revised their manuscript. This reviewer recommends to published in Molecules.

Reviewer 2 Report

The revised manuscript was significantly impouved and may be accepted for publication.